# Mitigating the Popularity Bias of Graph Collaborative Filtering: *A Dimensional Collapse Perspective*

**Yifei Zhang[†], Hao Zhu[‡,§], Yankai Chen[†], Zixing Song[†], Piotr Koniusz[*,§,‡], Irwin King[†]**

[†]The Chinese University of Hong Kong
[‡]Australian National University, [§]Data61♥CSIRO
{yfzhang, ykchen, zxsong, king}@cse.cuhk.edu.hk
allenhaozhu@gmail.com, piotr.koniusz@data61.csiro.au

## Abstract

Graph Collaborative Filtering (GCF) is widely used in personalized recommendation systems. However, GCF suffers from a fundamental problem where features tend to occupy the embedding space inefficiently (by spanning only a low-dimensional subspace). Such an effect is characterized in GCF by the embedding space being dominated by a few of popular items with the user embeddings highly concentrated around them. This enhances the so-called Matthew effect of the popularity bias where popular items are highly recommend whereas remaining items are ignored. In this paper, we analyze the above effect in GCF and reveal that the simplified graph convolution operation (typically used in GCF) shrinks the singular space of the feature matrix. As typical approaches (*i.e.*, optimizing the uniformity term) fail to prevent the embedding space degradation, we propose a decorrelation-enhanced GCF objective that promotes feature diversity by leveraging the so-called principle of redundancy reduction in embeddings. However, unlike conventional methods that use the Euclidean geometry to relax hard constraints for decorrelation, we exploit non-Euclidean geometry. Such a choice helps maintain the range space of the matrix and obtain small condition number, which prevents the embedding space degradation. Our method outperforms contrastive-based GCF models on several benchmark datasets and improves the performance for unpopular items.

## 1  Introduction

Collaborative Filtering (CF) has emerged as an effective technique for personalized recommendations [9, 6, 44, 8, 7]. By leveraging embedding and optimizing with a suitable loss function, CF can effectively capture complex patterns in user-item interactions [32, 16]. However, user-item interactions usually neglect the potential of the higher-hop connections. To this end, Graph Neural Networks (GNN) [36, 20, 14, 34, 50, 26] have been used to model high-hop connections in CF. Graph Collaborative Filtering (GCF) has also been proposed based on CF [15, 41].

Graph Contrastive Learning (GCL), a powerful tool for unsupervised learning, has recently been applied on a variety of different tasks [5, 47, 30, 49, 50]. Many methods also demonstrate that GCL improves the performance of GCF. The basic idea of GCL in GCF is to enhance training data with various graph augmentations and maximize the alignment between different data views [46, 43, 53]. However, the above methods ignore the disadvantages of GCL (*e.g.*, dimensional collapse) and the negative impact on user and item embeddings (*e.g.*, the Matthew effect) [4]. Dimensional Collapse (DC) [28] can be characterized by high similarity of features which are confined to a low-dimensional subspace or are contrated on curtain region of latent space( Figure 1b). DC limits the representational power of high-dimensional spaces by restricting the diversity of information that can be learned [28].

---

[*]The corresponding author.   Code: https://github.com/yifeiacc/LogDet4Rec/

37th Conference on Neural Information Processing Systems (NeurIPS 2023).

Although DC has been identified and tackled in a variety of domains [19], DC poses issues for GCL in the Graph Collaborative Filtering where it exacerbates the Matthew effect of popularity bias (as explained next).

A significant issue that sets GCL in GCF apart from other domains is the highly skewed data distribution over the graph where most users interact sporadically with a very small set of items within a vast interaction space comprising millions or even billions of items [4]. This results in extremely sparse user-item interactions and a long-tail item distribution following the Power Law. Propagating highly skewed data through the sparse bipartite graph via message passing enhances the Matthew effect of the popularity bias [4]. This means that several popular items may dominate the entire learning process, while other items are rarely recommended. Thus, a collapse mode may occur, where the user embedding collapses around several popular items, as shown in Figure 1a.

To migrate this issue, previous research argues that the so-called uniformity term included in the auxiliary contrastive loss can solve DC and so the uniformity term was adopted it in CF [45, 46, 38]. Particularly, Yu *et al.* [46] provide an empirical evidence that the features learned by the typical GCF model are non-uniformly distributed on the sphere, which causes the collapse. Wang *et al.* [38] have further investigated how optimizing the Bayesian Personalized Ranking (BPR) loss in CF can promote both the alignment and uniformity, and suggested to directly optimize them. However,

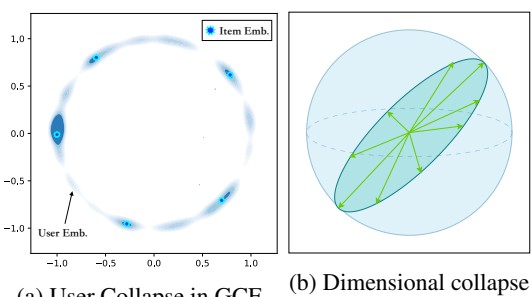

(a) User Collapse in GCF.   (b) Dimensional collapse in 3D hypersphere.

Figure 1

while optimizing the uniformity of both the user and item embeddings in CF can refine the embedding space, it does not guarantee the elimination of dimensional collapse. Thus, a fundamental question arises: is there any provable remedy for DC?

To this end, we analyze the connection between dimensional collapse and GCF by demonstrating that the simplified graph convolution operation for collaborative filtering does shrink the singular space of the feature matrix. Although previous methods promote the feature uniformity, it does not prevent the dimensional collapse. Thus, we propose a new objective, Decorrelation-Enhanced GCF, that employs geometry of the LogDet divergence to relax orthogonality constraints and encourage the feature diversity. Our method maintains the range space of the feature matrix with a small condition number (well-conditioned), providing theoretical guarantees against dimensional collapse. Our experimental results demonstrate the effectiveness of the proposed approach in addressing feature collapse. We outperform contrastive-based CF models on several benchmark datasets. Notably, our approach improves the performance for unpopular items.

We summarize our contributions below:

i. We demonstrate that embeddings learned from existing GCF models suffer from dimensional collapse. Our theoretical analysis reveals that the dimensional collapse in GCF arises from propagation, and that the current uniformity-based objective does not guarantee to alleviate it.

ii. To mitigate the issue of dimensional collapse, we propose a decorrelation-enhanced objective following the principle of redundancy reduction [47] and LogDet geometry.

iii. Instead of using the Frobenius norm (the Euclidean geometry) to relax the conditional optimization problem, we propose using the LogDet divergence, a Bregman divergence with theoretical guarantees, to preserve the range space of the covariance matrix.

Our experiments show that our proposed method effectively mitigates dimensional collapse and outperforms various contrastive learning-based CF models. Notably, it achieves improved the performance for unpopular items, thus it mitigates the Matthew effect.

## 2 Preliminaries

**Collaborative Filtering.**   Let $\mathcal{U}$ and $\mathcal{I}$ denote the user and item set, respectively. Given a set of observed user-item interactions $\mathcal{R} = \{(u, i) \mid u \text{ interacted with } i\}$, CF methods infer the score $s(u, i) \in \mathbb{R}$ for each unobserved user-item pair indicating how likely the user $u$ is to interact with

the item $i$. Then, items with the highest scores for each user will be recommended based on the predictions [32, 16]. Most CF methods use an encoder network $f(\cdot)$ that maps each user and item into a low-dimensional representation $f(u), f(i) \in \mathbb{R}^d$ (where $d$ is the dimensionality of the latent space). For example, the encoder in matrix factorization models is usually an embedding table, which directly maps each user and item to a latent vector based on their IDs. The encoder in graph-based models uses the neighborhood information. The predicted score is defined as the similarity between the user/item representations, *e.g.*, dot product, $s(u, i) = f(u)^\top f(i)$. Most studies adopt the pairwise BPR [31] loss to train the model:

$$\mathcal{L}_{bpr} = -\frac{1}{|\mathcal{R}|} \sum_{(u,i) \in \mathcal{R}} \log\left[\text{sigmoid}\left(s(u, i) - s\left(u, i^-\right)\right)\right], \tag{1}$$

where $i^-$ is a randomly sampled negative item that the user has not interacted with. Such a loss maximizes sigmoid scores: target user-item scores become higher than user-negative item scores.

**LightGCN.**   A linear graph neural network backbone, LightGCN [15], is one of the most successful backbones in Graph Collaborative Filtering. It leverages neighborhood information at different levels of locality, and combines outputs of different layers by the sum aggregation.

To facilitate the analysis, we present the matrix from of the LightGCN. Let $\mathbf{R} \in \mathbb{R}^{M \times N}$ be the matrix representation of set $\mathcal{R}$ where $R_{ui} = 1$ if $(u, i) \in \mathcal{R}$ (else $R_{ui} = 0$). Let $M$ and $N$ be the numbers of users and items, and $N' = M + N$. We then obtain the adjacency matrix of the user-item graph as:

$$\mathbf{A} = \begin{pmatrix} \mathbf{0} & \mathbf{R} \\ \mathbf{R}^\top & \mathbf{0} \end{pmatrix} \in \mathbb{R}^{N' \times N'}. \tag{2}$$

Let the embedding matrix of input layer be $\mathbf{E} \in \mathbb{R}^{N' \times d}$, where $d$ is the embedding size. Then the $K$-th layer of LightGCN in the matrix form is defined as:

$$\mathbf{Z} = f(\mathbf{E}; \tilde{\mathbf{A}}, K) = \alpha_1 \tilde{\mathbf{A}}\mathbf{E} + \alpha_2 \tilde{\mathbf{A}}^2 \mathbf{E} + \cdots + \alpha_K \tilde{\mathbf{A}}^K \mathbf{E} = \sum_{k=1}^{K} \alpha_k \tilde{\mathbf{A}}^k \mathbf{E}, \tag{3}$$

where $\alpha_1, \alpha_2, \ldots, \alpha_K$ are weights, and $\sum_{k=1}^{K} \alpha_k = 1$. Moreover, $\tilde{\mathbf{A}} = \mathbf{D}^{-\frac{1}{2}} \mathbf{A} \mathbf{D}^{-\frac{1}{2}}$ is a normalized adjacency matrix and the degree matrix $\mathbf{D}$ is an $N' \times N'$ diagonal matrix in which each entry $D_{ii}$ denotes the number of non-zero entries in the $i$-th row vector of the adjacency matrix $\mathbf{A}$.

**Dimensional Collapse (DC).**   Often referred to as spectral collapse [28], dimensional collapse is a prevalent phenomenon in representation learning. It occurs when the embedding space is dominated by a small number of large singular values, while the remaining singular values decay significantly as the training progresses. This phenomenon limits the representation power of high-dimensional spaces by restricting the diversity of information that can be learned.

## 3   Popularity Bias from a Perspective of Dimensional Collapse

### 3.1   Popularity Bias *vs*. Dimensional Collapse in Graph Collaborative Filtering

> *Dimensionality collapse* and *Popularity Bias* in CF are two sides of the same coin. The prevalence of popularity bias in GCF can be attributed to the significant imbalance within the data distribution across the recommendation bipartite graph [4]. This distribution often features numerous users engaging with a small subset of items, within a much larger interaction space encompassing millions or billions of items. Consequently, during the process of message passing within GCF, a few popular items can disproportionately influence learning, leaving out other items. This leads to the dimensional collapse in the embedding space where most user embeddings are around very few popular item embeddings.

We illustrate this by showing the 2D latent embedding space of the user and item in Figure 2:

i. Figure 2 (top) shows that both user and item embedding are clustered in a few regions, the gaps in the unpopular item embedding cause *effective rank* drop. We refer to this effect as the *dimensional collapse*. As the unpopular items are clustered in just one cluster, and the corresponding user cluster is non-dense so unpopular items do not get recommended often (*popularity bias*).

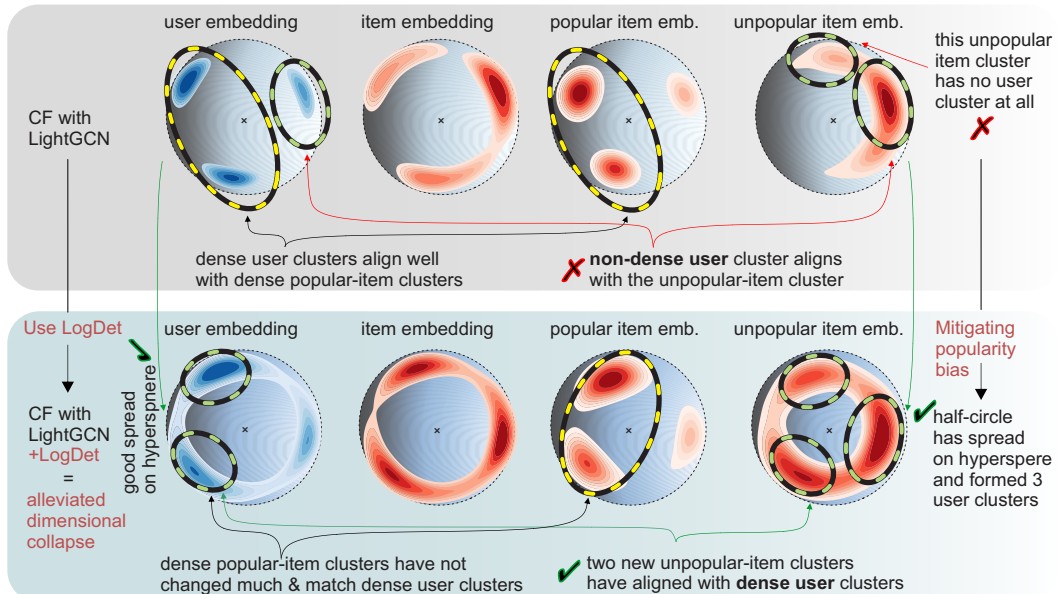

Figure 2: Mitigating Popualrity Bias. (*top*) We applied 2D PCA and the $\ell_2$ normalization on embeddings of LightGCN (Yelp2018 dataset). Notice that when only a non-dense user cluster aligns with the unpopular item cluster, these items will not be recommended frequently (✗). (*bottom*) With our LogDet-based loss, the unpopular-item cluster is spread into three clusters across surface of hypersphere. Newly formed clusters align well with dense user clusters (✓) so the unpopular items will be recommended more often. The dimensional Collapse manifests itself with gaps on the surface of hypersphere. Each plot was renormalized by its maximum density (darkest color). The 3D ball effect is to remind reader the embeddings exist on the surface hypersphere but in our simulation we use in fact a 2D sphere.

ii. If only the embeddings were spread uniformly on the surface of the hypersphere, it would align with dense user clusters which would reduce the popularity bias. Figure 2 (bottom) shows that thanks to more uniformly spread unpopular items, they form now 3 clusters, and 2 of them align well with dense user clusters, which means the popularity bias has been reduced.

### 3.2 Dimensional Collapse in Graph Collaborative Filtering

The previous section reveals that the occurrence of dimensional collapse is closely related to the presence of popularity bias, as depicted in Figure 2. Below, we show how the typical backbone of GCL in GCF (*i.e.*, LightGCN) potentially causes the DC in the learned embedding space, and why optimizing uniformity [38] (the best current solution) cannot guarantee to prevent DC.

**Lemma 1 (Equivalent reformulation of LightGCN.)** *The propagation process of LightGCN in Equation (3) can be viewed as a result of optimization of the following objective:*

$$\mathbf{Z}^* = \arg\min_{\mathbf{Z}} \ \|\mathbf{Z} - \tilde{\mathbf{A}}\mathbf{E}\|_F^2 + \eta \operatorname{tr}\left(\mathbf{Z}^\top \mathbf{L}\mathbf{Z}\right), \tag{4}$$

*where $\mathbf{L} = \mathbf{I} - \tilde{\mathbf{A}}$ is the normalized symmetric positive semi-definite graph Laplacian matrix.*
□ *Proof of Lemma 4 is in Appendix A.1.*

Lemma 1 shows that the propagation process of LightGCN can be derived from the optimization formulation which uses the Graph Laplacian regularization. This implies that the output embedding $\mathbf{Z}$ is smooth due to the Graph Laplacian regularization in Equation (4). Similar derivations for the graph neural network with learnable weights were shown by Ma *et al.* [29] and Zhu *et al.* [58].

To understand the dynamics of the embedding space, we denote the state of the embedding matrix and the output of the LightGCN at time $t$ as $\mathbf{E}(t)$ and the initial state $\mathbf{E}(0)$ is randomly initialized. Let $\{\sigma_n^{\mathbf{E}}\}_{n=1}^{N'}$ denote eigenvalues of $\mathbf{E}$ (descending order)

**Lemma 2 (Shrinking singular space of feature matrix.)** *Consider minimizing the objective in Lemma 1 with the Laplacian regularization term $\operatorname{tr}\left(\mathbf{Z}^\top \mathbf{L}\mathbf{Z}\right)$. The relative value of the ratio*

*of the smaller eigenvalues to the larger eigenvalues will decrease as $t$ increases. Formally,*
$\frac{\sigma_i^{\mathbf{E}}(t)}{\sigma_j^{\mathbf{E}}(t)} \leq \frac{\sigma_i^{\mathbf{E}}(t')}{\sigma_j^{\mathbf{E}}(t')}$, $\forall t < t'$, $i > j$ *(we ignore the trivial case $i = j$). Then:*

$$\lim_{t \to \infty} \frac{\sigma_i^{\mathbf{E}}(t)}{\sigma_j^{\mathbf{E}}(t)} = 0, \quad \forall i > j \quad such\ that \quad \sigma_i^{\mathbf{E}} < \sigma_j^{\mathbf{E}}. \tag{5}$$

□ Proof of Lemma 2 is in Appendix A.2.

> Based on Lemmas 1 and 2, we conclude that due to the propagation step in Equation (3), the embedding space of feature matrix is dominated by few largest eigenvalues. This is a poof of dimensional collapse of GCF pipelines which are typically based on the LightGCN backbone.

### 3.3 Optimizing Uniformity in Graph Collaborative Filtering.

Typical contrastive GCF approaches adopt the BPR loss from Equation (1) and the InfoNEC loss. Wang *et al.* [38] argued that the so-called alignment and uniformity [38] loss terms contribute to higher recommendation performance:

$$\mathcal{L}_{align} = \mathbb{E}_{(u,i) \sim \mathcal{R}} \left[ \| f_\theta(u) - f_\theta(i) \|_2^2 \right], \quad \text{(alignment)}$$
$$\mathcal{L}_{uni} = -\log \left[ \mathbb{E}_{u,u^- \sim \mathcal{U}} e^{(2 - 2 f_\theta(u)^\top f_\theta(u^-))} \right] - \log \left[ \mathbb{E}_{i,i^- \sim \mathcal{I}} e^{(2 - 2 f_\theta(i)^\top f_\theta(i^-))} \right], \quad \text{(uniformity)} \tag{6}$$

where $\mathcal{L}_{algin}$ captures the user-item relations. $\mathcal{L}_{uni}$ encourages embeddings to be distributed evenly on the surface of $\ell_2$ ball. However, $\mathcal{L}_{uni}$ does not avoid the dimensional collapse shown in Figure 1.

Concluding, Contrastive GCF suffers from several problems:

i. DC [17] is characterized by one or more singular values in the feature matrix to be zero, *e.g.*, $\lambda_i = 0$. $\mathcal{L}_{uni}$ may yield a finite reward (decrease of $\mathcal{L}_{uni}$) to avoid the dimensional collapse ($\lambda_i = 0$). However, if the remaining loss terms yield a higher reward (decrease in their value) than $\mathcal{L}_{uni}$ for occurrence of the dimensional collapse ($\lambda_i = 0$) then DC will occur.

ii. As temperature $1/\tau$ increases, $\mathcal{L}_{uni}$ degrades to a hard minimum. Take just the user part of Equation (6), include the temperature variable $1/\tau$, and notice that:

$$\lim_{\tau \to 0} -\tau \log \left[ \mathbb{E}_{u,u^- \sim \mathcal{U}} e^{(2 - 2 f_\theta(u)^\top f_\theta(u^-))/\tau} \right] = \min_{(u,u^-) \in \mathcal{U} \times \mathcal{U}} 2 - f_\theta(u)^\top f_\theta(u^-), \tag{7}$$

which explains the empirical observation of Wang and Liu [39] that the uniformity of embeddings worsens as the temperature increases.

iii. As Lemma 2 shows, LightGCN itself is also prone to suffering from the dimensional collapse.

Thus, in what follows we consider the redundancy reduction principle to mitigate the above issues.

## 4 Methodology

**Graph Collaborative Filtering with Decorrelation.** From the perspective of redundancy reduction [47, 2], achieving high-quality self-supervised embeddings requires that: (i) positive pairs exhibit similar semantics; (ii) the embeddings follow a non-trivial constant distribution, with a trivial distribution indicating that all embeddings collapse into a single point; and (iii) there is zero correlation between different features.

In the context of collaborative filtering, we can frame this problem as follows:

$$\min_\theta \ \mathcal{L}_{align}(\theta) \tag{8}$$
$$\text{s.t.} \ \sum_{i \in \mathcal{U}} f_\theta(u) f_\theta(u)^\top = \sum_{i \in \mathcal{I}} f_\theta(i) f_\theta(i)^\top = \mathbf{I},$$

Table 1: Several decorrelation terms, properties and teaser results (Yelp2018).

|  | $\mathcal{L}_{uni}$ | $\mathcal{L}_{soft}$ | $\mathcal{L}_{logdet}$ | Stiefel |
|---|---|---|---|---|
| **isotropic** | soft | soft | soft | hard |
| **full-rank** | soft | soft | hard | hard |
| **recall@20 (%)** | 7.12 | 7.09 | **7.25** | 7.03 |

where $\mathcal{L}_{align}(\theta) = \mathbb{E}_{(i,u) \sim \mathcal{R}} \left[ \| f_\theta(u) - f_\theta(i) \|_2^2 \right]$ (from Equation (6)) captures user-item preferences. Constraints $\sum_{i \in \mathcal{U}} f_\theta(u) f_\theta(u)^\top = \sum_{i \in \mathcal{I}} f_\theta(i) f_\theta(i)^\top = \mathbf{I}$ help embeddings follow the isotropic Normal distribution. The isotropy prevents the dimensional collapse but its decorrelating effect may

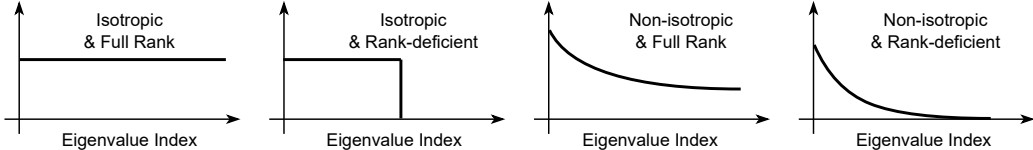

Figure 3: The spectrum (eigenvalue distribution) of the covariance matrix under various loss penalties. Various penalties realize some composition of {isotropic, non-isotropic}×{full rank, rank-defficient}, *e.g.*, our LogDet formulation promotes the (isotropic, full rank) case.

be too strong. Equation (8) may be solved by applying Lagrangian multipliers or the Stiefel manifold (GeoTorch [25]) or by converting hard constraints into soft constraints:

$$\min_\theta \ \mathcal{L}_{align}(\theta) + \lambda \cdot \mathcal{L}_{soft} \quad \text{where} \quad \mathcal{L}_{soft} = \Big\| \sum_{i \in \mathcal{U}} f_\theta(u) f_\theta(u)^\top - \mathbf{I} \Big\|_F^2 + \Big\| \sum_{i \in \mathcal{I}} f_\theta(i) f_\theta(i)^\top - \mathbf{I} \Big\|_F^2, \quad (9)$$

However, relaxation $\mathcal{L}_{soft}$ in Equation (9) suffers from the same problem as $\mathcal{L}_{uni}$ in Equation (6). Specifically, these both regularization terms yield a finite penalty for any singular values $\lambda_i = 0$ leading to dimensional collapse if other loss terms yield a higher reward than the penalty is. Thus, we explore the application of Bregman divergence. Our study indicates that the proposed approach preserves the range space of the covariance matrix, promoting a full-rank embedding matrix.

**Decorrelation by the LogDet Divergence.** Figure 3 and Table 1 summarizes properties of different decorrelation terms. While $\mathcal{L}_{uni}$ and $\mathcal{L}_{soft}$ encourage soft decorrelation (approximate isotropy), they may fail to impose full rank on the embedding matrix. Stiefel-based optimization enjoys hard constraints but full decorrelation (exact isotropy) may be too restrictive. In contrast, LogDet divergence based $\mathcal{L}_{logdet}$ encourages isotropy in the soft manner and imposes full rank on embedding matrix as shown below. Firstly, we review the Bregman divergence and introduce our LogDet based decorrelation penalty.

**Definition 1 (Bregman Divergence on Matrices.)** *Let $\mathbf{X}, \mathbf{Y} \in \mathcal{S}^d$ be two symmetric matrices. Let $\phi : \mathcal{S}^n \to \mathbb{R}$ be a strictly convex and differentiable matrix function. The Bregman matrix divergence is defined as:*
$$D_\phi(\mathbf{X}, \mathbf{Y}) = \phi(\mathbf{X}) - \phi(\mathbf{Y}) - \text{tr}\left((\nabla\phi(\mathbf{Y}))^\top (\mathbf{X} - \mathbf{Y})\right). \quad (10)$$
The Bregman divergence is a measure of the nearness between matrices $\mathbf{X}$ and $\mathbf{Y}$. It is computed by using the first-order Taylor approximation of a convex generating function $\phi(\cdot)$. The Bregman divergence is always non-negative, and it becomes zero only when the two matrices are equal.

As $\mathbf{\Sigma}_\mathcal{U} = \sum_{u \in \mathcal{U}} f_\theta(u) f_\theta(u)^\top \in \mathcal{S}_+^d$ and $\mathbf{\Sigma}_\mathcal{I} = \sum_{i \in \mathcal{I}} f_\theta(i) f_\theta(i)^\top \in \mathcal{S}_+^d$ are symmetric, and at least positive semi-definite (PSD) matrices but we desire them to be strictly positive definite (PD) matrices ($\mathcal{S}_{++}^d$) to prevent the dimensional collapse, we firstly generalize $\mathcal{L}_{soft}$ in Equation (9) to its Bregman divergence based variant:
$$\mathcal{L}_{breg} = D_\phi(\mathbf{\Sigma}_\mathcal{U}, \mathbf{I}) + D_\phi(\mathbf{\Sigma}_\mathcal{I}, \mathbf{I}). \quad (11)$$

Different $\phi(\cdot)$ lead to different measures of nearness. For example, $\phi_F(\mathbf{X}) = \|\mathbf{X}\|_F^2$ yields the Frobenius-based $D_{\phi_F}(\mathbf{\Sigma}_\mathcal{U}, \mathbf{I}) + D_{\phi_F}(\mathbf{\Sigma}_\mathcal{I}, \mathbf{I}) = \|\mathbf{\Sigma}_\mathcal{U} - \mathbf{I}\|_F^2 + \|\mathbf{\Sigma}_\mathcal{I} - \mathbf{I}\|_F^2$ penalty $\mathcal{L}_{soft}$ in Equation (9).

Let $\mathbf{X} = \mathbf{U}\mathbf{\Lambda}\mathbf{U}^\top$ be an eigenvalue decomposition of $\mathbf{X}$, and $\mathbf{\Lambda} = \text{diag}([\lambda_1, \dots, \lambda_d])$. Choosing $\phi_{ld}(\mathbf{X}) = -\log\det\mathbf{X} = -\sum_i \log \lambda_i$ leads to the LogDet divergence $D_{\phi_{ld}}(\mathbf{X}, \mathbf{Y}) = \text{tr}\left(\mathbf{X}\mathbf{Y}^{-1}\right) - \log\det\left(\mathbf{X}\mathbf{Y}^{-1}\right) - d$ where $\mathbf{X}, \mathbf{Y} \in \mathcal{S}_{++}^d$. Then generalizing $\mathcal{L}_{soft}$ in Equation (9) to its LogDet divergence variant by computing $D_{\phi_{ld}}(\mathbf{\Sigma}_\mathcal{U}, \mathbf{I}) + D_{\phi_{ld}}(\mathbf{\Sigma}_\mathcal{I}, \mathbf{I})$ and removing $-2d$ constant yields:

$$\mathcal{L}_{logdet} = \text{tr}\left(\mathbf{\Sigma}_\mathcal{U} + \mathbf{\Sigma}_\mathcal{I}\right) - \log\det\left(\mathbf{\Sigma}_\mathcal{U}\mathbf{\Sigma}_\mathcal{I}\right) - 2d. \quad (12)$$

---

Our LogDet decorrelation-enhanced GCF objective becomes:
$$\mathcal{L} = \mathbb{E}_{(i,u) \sim \mathcal{R}} \left[\|f_\theta(u) - f_\theta(i)\|_2^2\right] + \lambda \cdot \left(\text{tr}\left(\mathbf{\Sigma}_\mathcal{U} + \mathbf{\Sigma}_\mathcal{I}\right) - \log\det\left(\mathbf{\Sigma}_\mathcal{U}\mathbf{\Sigma}_\mathcal{I}\right)\right). \quad (13)$$

In contrast to the Frobenius-based $\mathcal{L}_{soft}$, the LogDet-based penalty $\mathcal{L}_{logdet}$ maintains the range space of the covariance matrices and upper bounds the condition number of the matrices. This ensures the user/item embedding matrices are of full rank which prevents the dimensional collapse.

# 5 Theoretical Analysis.

To use the LogDet divergence based penalty $\mathcal{L}_{logdet}$ in GCF, we discuss important to our work theoretical properties of the LogDet divergence and $\mathcal{L}_{logdet}$ for rank-deficient matrices.

**LogDet Divergence and $\mathcal{L}_{logdet}$ for Rank-deficient Matrices.**

**Corollary 1.** *LogDet divergence $D_{\phi_{ld}}(\mathbf{X}, \mathbf{Y})$ is finite iff* $\text{rank}(\mathbf{X}) = \text{rank}(\mathbf{Y})$ *[23], which implies that $\mathcal{L}_{logdet} = D_{\phi_{ld}}(\mathbf{\Sigma}_{\mathcal{U}}, \mathbf{I}) + D_{\phi_{ld}}(\mathbf{\Sigma}_{\mathcal{I}}, \mathbf{I}) = \infty$ if $\text{rank}(\mathbf{\Sigma}_{\mathcal{U}}) < \text{rank}(\mathbf{I}) = d \ \lor \ \text{rank}(\mathbf{\Sigma}_{\mathcal{I}}) < \text{rank}(\mathbf{I}) = d$.*

**Corollary 2.** *Let $\mathbf{\Sigma}_{\mathcal{U}}, \mathbf{\Sigma}_{\mathcal{I}} \in \mathcal{S}_{++}^d$. Note that $\mathcal{L}_{soft} = D_{\phi_F}(\mathbf{\Sigma}_{\mathcal{U}}, \mathbf{I}) + D_{\phi_F}(\mathbf{\Sigma}_{\mathcal{I}}, \mathbf{I})$ is finite if both $\|\mathbf{\Sigma}_{\mathcal{U}}\|_2$ and $\|\mathbf{\Sigma}_{\mathcal{I}}\|_2$ are finite. There exist cases where some infinitesimal change $\Delta\mathbf{\Sigma}_{\mathcal{U}}$ and/or $\Delta\mathbf{\Sigma}_{\mathcal{I}}$ will make $\mathbf{\Sigma}_{\mathcal{U}} + \Delta\mathbf{\Sigma}_{\mathcal{U}}$ and/or $\mathbf{\Sigma}_{\mathcal{I}} + \Delta\mathbf{\Sigma}_{\mathcal{I}}$ positive semi-definite (PSD). Define the gain of the Bregman divergence based decorrelation penalty as $g_\phi(\mathcal{L}_{reg}) = D_\phi(\mathbf{\Sigma}_{\mathcal{U}}, \mathbf{I}) + D_\phi(\mathbf{\Sigma}_{\mathcal{I}}, \mathbf{I}) - [D_\phi(\mathbf{\Sigma}_{\mathcal{U}} + \Delta\mathbf{\Sigma}_{\mathcal{U}}, \mathbf{I}) + D_\phi(\mathbf{\Sigma}_{\mathcal{I}} + \Delta\mathbf{\Sigma}_{\mathcal{I}}, \mathbf{I})]$. Notice, in case of change from PD to SPD matrix, gain $g_\phi(\mathcal{L}_{reg}) < 0$. Let the gain of the alignment loss be $g(\mathcal{L}_{align}) \ll \infty$. Then one may easily construct examples where $g(\mathcal{L}_{align}) - g_{\phi_F}(\mathcal{L}_{soft}) > 0$. That is to say, the net loss $-g_{\phi_F}(\mathcal{L}_{soft})$ in case of covariance degradation to SPD is lesser than the net gain $g(\mathcal{L}_{align})$ due to increased user-item alignment. In contrast, from Corollary 1, $g(\mathcal{L}_{align}) - g_{\phi_{ld}}(\mathcal{L}_{logdet}) = -\infty$ in case of covariance degradation to SPD, which prevents the dimensional collapse.*
☐ *Proof of Corollary 2 is in Appendix A.3.*

**Corollary 3.** *Expression $-\log\det(\mathbf{\Sigma}_{\mathcal{U}}\mathbf{\Sigma}_{\mathcal{I}})$ from Equation (12) might fail to yield $\infty$ if $\mathbf{\Sigma}_{\mathcal{U}}\mathbf{\Sigma}_{\mathcal{I}}$ is indefinite. However, covariance matrices are at least SPD by definition, and so is their matrix product.*

> Corollaries 1, 2 and 3 demonstrate that $\mathcal{L}_{logdet}$ naturally maintains the full-rank constraints on covariance matrices of user and item embeddings.

**Minimizing $D_{\phi_{ld}}(\mathbf{\Sigma}, \mathbf{I})$ Reduces the Condition Number of $\mathbf{\Sigma}$.** Feature decorrelation is closely related to the condition number. If the spectrum of a matrix is dominated by the leading eigenvalue, the matrix is ill-conditioned and has a large condition number. Let $\{\lambda_i\}_{i=0}^d$ be the eigenvalues of the matrix $\mathbf{\Sigma}$. The condition number is defined as $\text{cond}(\mathbf{\Sigma}) = \frac{\lambda_{\max}}{\lambda_{\min}}$. The ratio between the largest eigenvalue and the smallest eigenvalue quantifies the flatness (isotropy) of the spectrum. For example, $\text{cond}(\mathbf{\Sigma}) = 1$ implies a fully balanced spectrum with $\lambda_i = \lambda_j$ for all $i \neq j$ (*e.g.*, spectrum balancing [21, 22, 48, 52]). Below we show that our proposed method minimizes the upper bound of $\text{cond}(\mathbf{\Sigma})$.

**Lemma 3.** *The condition number of $\mathbf{\Sigma}$, $\text{cond}(\mathbf{\Sigma}) = \frac{\lambda_{\max}}{\lambda_{\min}}$, is upper bounded by $D_{\phi_{ld}}(\mathbf{\Sigma}, \mathbf{I})$ as:*

$$\text{cond}(\mathbf{\Sigma}) \leq 4\exp\left(D_{\phi_{ld}}(\mathbf{\Sigma}, \mathbf{I})\right). \tag{14}$$

☐ *Proof of Lemma 3 is in Appendix A.4.*

> Lemma 3 implies that the sum of condition numbers of $\mathbf{\Sigma}_{\mathcal{U}}$ and $\mathbf{\Sigma}_{\mathcal{I}}$ is upper bounded by the LogDet divergence based penalty $\mathcal{L}_{logdet}$. Minimizing that penalty implies the flattening of spectrum implies decorrelation/redundancy reduction of embedding.

# 6 Experiments

**Datasets.** To evaluate the effectiveness of our GCF with the LogDet divergence based penalty $\mathcal{L}_{logdet}$, denoted as $\text{GCF}_{logdet}$, we conduct experiments on three large-scale public datasets: Yelp2018 [15], Amazon-kindle [43], and Alibaba-iFashion [43]. See statistics of these datasets in Appendix A.6. The datasets were split into the training, validation, and test set, with the ratio of $7:1:2$. As per the methodology suggested by [43, 15], we first identified the best hyperparameters on the validation set and then merged the training and validation sets to train the model. Finally, the test set was used to evaluate the model using the relevancy-based metric Recall@20 and the ranking-aware metric NDCG@20.

**Baselines.** Apart from the typical GCF models, *i.e.*, **NGCF** [41] and **LightGCN** [15], we also compare $\text{GCF}_{logdet}$ with recent data CL-based recommendation models. **SGL** [43] and DNN+SSL [45] adopt CL as an auxiliary task and conduct feature masking for contrastive learning (CL). **BUIR**

Table 2: Performance comparison for different contrastive GCF model that optimize uniformity.

| Methods | | Yelp2018 | | Amazon-Kindle | | Alibaba-iFashion | |
|---|---|---|---|---|---|---|---|
| | | Recall@20 | NDCG@20 | Recall@20 | NDCG@20 | Recall@20 | NDCG@20 |
| **1 layer** | LightGCN | 0.0590 | 0.0484 | 0.1871 | 0.1186 | 0.0845 | 0.0390 |
| | SGL-ED | 0.0637 | 0.0526 | 0.1936 | 0.1231 | 0.0932 | 0.0442 |
| | SimGCL | 0.0689 | 0.0572 | 0.2087 | **0.1361** | 0.1036 | 0.0505 |
| | DirectAU | 0.0674 | 0.0522 | 0.1971 | 0.1239 | 0.0969 | 0.0426 |
| | GCF$_{logdet}$ | **0.0694** | **0.0577** | **0.2103** | 0.1321 | **0.1060** | **0.0510** |
| **2 layers** | LightGCN | 0.0622 | 0.0504 | 0.2033 | 0.1284 | 0.1053 | 0.0505 |
| | SGL-ED | 0.0668 | 0.0549 | 0.2084 | 0.1341 | 0.1062 | 0.0514 |
| | SimGCL | 0.0719 | 0.0601 | 0.2071 | 0.1341 | 0.1119 | 0.0548 |
| | DirectAU | 0.0702 | 0.0584 | 0.2034 | 0.1352 | 0.1043 | 0.0549 |
| | GCF$_{logdet}$ | **0.0720** | **0.0607** | **0.2143** | **0.1381** | **0.1180** | **0.0557** |
| **3 layers** | LightGCN | 0.0639 | 0.0525 | 0.2057 | 0.1315 | 0.0955 | 0.0461 |
| | SGL-ED | 0.0675 | 0.0555 | 0.2090 | 0.1352 | 0.1093 | 0.0531 |
| | SimGCL | 0.0721 | 0.0601 | 0.2104 | 0.1374 | 0.1151 | 0.0567 |
| | DirectAU | 0.0713 | 0.0602 | 0.2047 | 0.1385 | 0.1136 | 0.0586 |
| | GCF$_{logdet}$ | **0.0725** | **0.0612** | **0.2151** | **0.1418** | **0.1210** | **0.0601** |

Table 3: Performance comparison with other models.

| Method | Yelp2018 | | Amazon-Kindle | | Alibaba-iFashion | |
|---|---|---|---|---|---|---|
| | Recall@20 | NDCG@20 | Recall@20 | NDCG@20 | Recall@20 | NDCG@20 |
| NGCF | 0.0579 | 0.0477 | 0.1954 | 0.1263 | 0.1043 | 0.0486 |
| LightGCN | 0.0639 | 0.0525 | 0.2057 | 0.1315 | 0.1053 | 0.0505 |
| NCL | 0.0670 | 0.0562 | 0.2090 | 0.1348 | 0.1088 | 0.0528 |
| BUIR | 0.0487 | 0.0404 | 0.0922 | 0.0528 | 0.0830 | 0.0384 |
| DNN+SSL | 0.0483 | 0.0382 | 0.1520 | 0.0989 | 0.0818 | 0.0375 |
| MixGCF | 0.0713 | 0.0589 | 0.2098 | 0.1355 | 0.1085 | 0.0520 |
| SGL-ED | 0.0675 | 0.0555 | 0.2090 | 0.1352 | 0.1093 | 0.0531 |
| SimGCL | 0.0721 | 0.0601 | 0.2104 | 0.1374 | 0.1151 | 0.0567 |
| DirectAU | 0.0713 | 0.0602 | 0.2047 | 0.1385 | 0.1136 | 0.0586 |
| GCF$_{logdet}$ | **0.0725** | **0.0612** | **0.2151** | **0.1418** | **0.1210** | **0.0601** |

[24] has a two-branch architecture with a target network and an online network, utilizing only positive examples for self-supervised recommendation. **MixGCL** [18] introduces the hop mixing technique for synthesizing hard negatives for GCF through embedding interpolation. **NCL** [27] is a recent contrastive model that designs a prototypical contrastive objective to capture the correlations between a user/item and its context. **SimGCL** [46] is a CL-based model for recommendation that does not rely on graph augmentation. **DirectAU** [38] proposes optimizing the alignment and uniformity of both user and item directly for recommendation.

**Implementation.** We referred to the best hyperparameter settings reported in the original papers of the baselines and then fine-tuned them with the grid search. Detailed settings are in Appendix A.5.

## 6.1 Main Results

**Decorrelation *vs*. Uniformity.** Within this section, we compare the performance of our proposed method, GCF$_{logdet}$, with contrastive-based methods (*i.e.*, SGL, DirectAU) that optimize uniformity to prevent the dimensional collapse in CF. Furthermore, we demonstrate the effectiveness of our method by varying the number of encoder layers. Table 3 shows the performance of different baseline CF methods and our GCF$_{logdet}$. In the vast majority of cases, contrastive-based methods (*e.g.*, SGL, DirectAU, SimGCL) significantly outperform LightGCN. The largest improvements are observed on Alibaba-iFashion, where the performance of GCF$_{logdet}$ surpasses that of LightGCN under 1 and 3 layers settings. This observation supports our conclusion that LightGCN is susceptible to the dimensional collapse, leading to poor results. Additionally, we observe that DirectAU outperforms SGL, indicating that graph augmentation is not the primary factor for enhancing the performance of the CL-based recommendation model. Instead, our findings demonstrate the importance of designing appropriate loss functions over sophisticated augmentation. Finally, GCF$_{logdet}$ achieves the best results on the all three datasets. Specifically, GCF$_{logdet}$ outperforms DirectAU and SGL, highlighting that decorrelation is more effective than optimizing uniformity for preventing dimensional collapse.

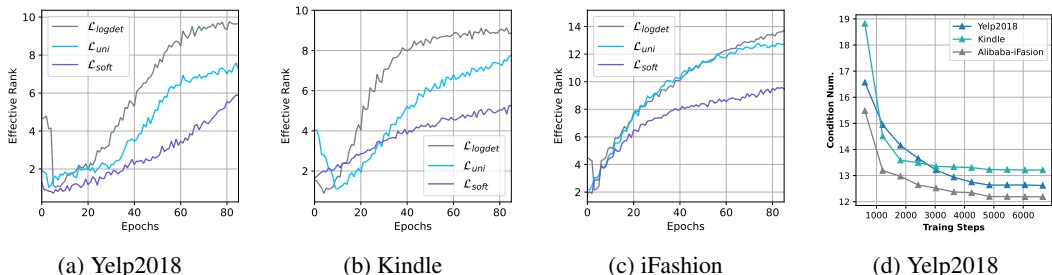

|             |             | (a) Yelp2018 | (b) Kindle | (c) iFashion | (d) Yelp2018 |

Figure 4: The effective rank (higher is better) w.r.t. epoch number and condition number in Yelp2018.

Table 4: Performance comparison with other models.

| Method | | Yelp2018 | | Amazon-Kindle | | iFashion | |
|---|---|---|---|---|---|---|---|
| Encoder | Loss | Recall@20 | NDCG@20 | Recall@20 | NDCG@20 | Recall@20 | NDCG@20 |
| LightGCN | $\mathcal{L}_{uni}$ | 0.0713 | 0.0602 | 0.2107 | 0.1385 | 0.1136 | 0.0586 |
| | $\mathcal{L}_{soft}$ | 0.0709 | 0.0592 | 0.1982 | 0.1351 | 0.1112 | 0.0580 |
| | Stiefel | 0.0703 | 0.0582 | 0.1882 | 0.1352 | 0.1071 | 0.0528 |
| | $\mathcal{L}_{logdet}$ | **0.0725** | **0.0612** | **0.2151** | **0.1418** | **0.1210** | **0.0601** |
| MLP | $\mathcal{L}_{uni}$ | 0.0702 | 0.0589 | 0.1960 | 0.1325 | 0.1085 | 0.0580 |
| | $\mathcal{L}_{soft}$ | 0.0699 | 0.0582 | 0.2060 | 0.0989 | 0.1018 | 0.0595 |
| | Stiefel | 0.0691 | 0.0579 | 0.1852 | 0.1191 | 0.0958 | 0.0523 |
| | $\mathcal{L}_{logdet}$ | **0.0721** | **0.0601** | **0.2154** | **0.1414** | **0.1151** | **0.0607** |

**Performance Comparison with the State of the Art.** To further demonstrate the exceptional performance of GCF$_{logdet}$, we conduct a comparative analysis with several recently proposed recommendation models that are based on augmentation or contrastive learning techniques. As illustrated in Table 3, GCF$_{logdet}$ outperforms the other models by a significant margin, achieving the best performances, respectively. Additionally, NCL and MixGCF, which utilize LightGCN as their backbone, exhibit competitive performance. In contrast, DNN+SSL and BUIR fall short of expectations and are not comparable to LightGCN. We attribute their suboptimal performance to the fact that DNNs have been proven to be effective only when abundant user/item features are available. In our datasets, however, such features are unavailable, and self-supervision signals are created by masking item embeddings, which makes it difficult for these models to perform well in this context. We also report performance on Recall@K and NDCG@K with $K = 5, 10$ in Table 9.

## 6.2 Quantitative Analysis

**Measuring the Dimensional Collapse.** The evidence of dimensional collapse was identified in contrastive learning by observing the spectrum of representations [19]. The rank of the matrix corresponds to the number of dimensions retained by the transformation (*i.e.*, the dimension of its range) but reveals nothing about the shape of the spectrum. The effective rank [33] (see the definition in the Appendix A.7) quantifies how balanced the spectrum is by means of the spectral entropy. Thus, to show the dimensional collapse tendency when training the model, instead of showing the distribution of eigenvalues, we use the effective rank. Figure 4 shows that the effective rank increases as the training progresses, indicating an increasingly balanced eigenvalue distribution. GCF$_{logdet}$ consistently achieves higher effective rank compared to other methods. This finding not only verifies the existence of the dimensional collapse in CF but also indicates the effectiveness of our GCF$_{logdet}$. Figure 4d further shows that the condition number decreases as more batches are processed, thus balancing the spectrum of embedding matrices.

**Overcoming the Popularity Bias.** Below we investigate whether our GCF$_{logdet}$ can reduce the popularity bias by promoting more uniform representations and alleviating the problem of dimensional collapse. We partition the test set into three subsets based on item popularity. Specifically, we label 80% of the items with the fewest clicks/purchases as "Unpopular," 5% of the most clicked/purchased items as "Popular," and the remaining items as "Normal." We then conduct experiments to measure Recall@20 contributed by each group, with the overall Recall@20 value being the sum of the values

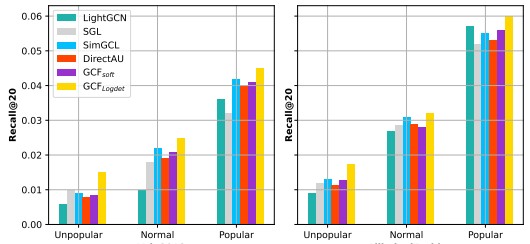

Figure 5: LogDet *vs*. other baselines. LightGCN is the backbone used in all methods.

Table 5: The numerical relative improvement of recall@20 in Figure 5 over LightGCN.

| Model | Yelp2018 | | | iFasion | | |
|---|---|---|---|---|---|---|
| | Unpop. | Norm. | Pop. | Unpop. | Norm. | Pop. |
| SGL | 0.63 | 0.83 | -0.11 | 0.33 | 0.05 | -0.08 |
| SimGCL | 0.52 | 1.21 | 0.16 | 0.44 | 0.14 | -0.03 |
| DirectAU | 0.33 | 0.91 | 0.11 | 0.27 | 0.07 | -0.07 |
| $GCF_{soft}$ | 0.41 | 1.10 | 0.13 | 0.41 | 0.03 | -0.01 |
| $GCF_{logdet}$ | **1.51** | **1.48** | **0.25** | **0.94** | **0.18** | **0.05** |

from all three groups. The results are presented in Figure 5 and Table 5, which show that our method significantly improves the ability to recommend unpopular items. In contrast, LightGCN tends to recommend popular items and achieves the highest recall value on Normal and Popular items.

**Ablations on Different Decorrelation Penalties.** Table 4 compares the performance on LightGCN and MLP equipped with different decorrelation penalties. $\mathcal{L}_{logdet}$ appears to consistently outperform other penalties, thus verifying its ability to deal with the dimensional collapse and balancing the spectrum of embeddings. Appendix A.8 compares the KL Matrix Divergence and the Riemannian metric in terms of speed. Appendix A.9 compares different distance types.

## 7 Related Works

**Graph Contrastive Learning** applies CL to the graph domain. By adapting DeepInfoMax [1] to graph representation learning, DGI [37] learns embedding by maximizing the mutual information to discriminate between nodes of original and corrupted graphs. REFINE [54] uses a simple negative sampling term inspired by skip-gram models. COLES [57] and GLEN [56] link the GCL to the form of Laplacian Eigenmap with negative sampling, extended also to image domain by EASE [55]. GRACE [59]/GraphCL [13] create views via graph augmentation for node/graph-level task. COSTA [51] and SFA [52] create graph views via feature augmentations.

**Graph Collaborative Filtering.** Graph Neural Networks (GNNs) have proven to be powerful architectures for modeling recommendation data, replacing MLP-based models and improving the performance of neural recommender systems [20, 15]. GNNs are a foundation for several state-of-the-art recommendation models. The most common GNN variant is GCN, which has driven the development of graph neural recommendation models, such as GCMC, NGCF, and LightGCN [3, 41, 15]. These GCN-based models refine embeddings and perform graph reasoning by gathering neighborhood information in the user-item graph. Among such models, LightGCN stands out for its effectiveness and simplicity by eliminating transformation matrices and non-linear activation functions. The success of LightGCN has inspired subsequent CL-based recommendation models, such as SGL and SimGCL [43, 46].

**Graph Contrastive Learning for Recommendation.** Inspired by the success of CL in other fields [11, 5, 35], CL has been combined with recommendation systems. Yao *et al*. [45] proposed a feature dropout based two-tower architecture for large-scale item recommendation. NCL [27] designed a prototypical contrastive objective to capture the correlations between a user/item and its context. SimGCL [46] adds noise to the embedding space to perform feature augmentation to enhance the performance of recommendations. DirectAU [38] proposes optimizing the alignment and uniformity of both user and item during training. The most widely used model, SGL [43], applies edge/node dropout to augment the graph data. Although these methods have demonstrated their effectiveness, they pay little attention to why CL can enhance recommendations.

## 8 Conclusions

We have investigated the popularity bias in Graph Collaborative Filtering, and showed it results from the dimensional collapse of the embedding space. We have demonstrated that the commonly used backbone in GCF, LightGCN, is prone to pose the dimensional collapse. To this end, we have proposed the LogDet divergence based decorrelation penalty for the GCF. We have showed that $GCF_{logdet}$ promotes full-rank embedding matrices and yields small condition number with a theoretical guarantee. For these reasons, $GCF_{logdet}$ has outperformed other contrastive-based GCF models on several benchmark datasets and improved the performance for unpopular items.

## Acknowledgments

The work described in this paper was partially supported by the National Key Research and Development Program of China (No. 2018AAA0100204), RGC Research Impact Fund (RIF), R5034-18 (CUHK 2410021), and RGC General Research Funding Scheme (GRF) 14222922 (CUHK 2151185). PK is supported by CSIRO's Science Digital.

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

# Mitigating the Popularity Bias of Graph Collaborative Filtering: *A Dimensional Collapse Perspective* (*Supplementary Material*)

**Yifei Zhang**[†], **Hao Zhu**[‡,§], **Yankai Chen**[†], **Zixing Song**[†], **Piotr Koniusz**[*,§,‡], **Irwin King**[†]
[†]The Chinese University of Hong Kong
[‡]Australian National University, [§]Data61♥CSIRO
{yfzhang, ykchen, zxsong, king}@cse.cuhk.edu.hk
allenhaozhu@gmail.com, piotr.koniusz@data61.csiro.au

## A  Appendices

### A.1  Proof of Lemma 1

**Proof 1** *One can set derivative of Equation (3) with respect to $\mathbf{Z}$ to zero and get the optimal $\mathbf{Z}$ as:*

$$\frac{\partial \left[ \|\mathbf{Z} - \tilde{\mathbf{A}}\mathbf{E}\|_F^2 + \eta \operatorname{tr}\left(\mathbf{Z}^T \mathbf{L} \mathbf{Z}\right) \right]}{\partial \mathbf{Z}} = 0 \quad \Rightarrow \quad \mathbf{Z} - \tilde{\mathbf{A}}\mathbf{E} + \eta \mathbf{L}\mathbf{Z} = 0. \tag{15}$$

*Note that $\det(\mathbf{I} + \eta\mathbf{L}) > 0$, thus matrix $\{\mathbf{I} + \eta\mathbf{L}\}^{-1}$ exists. Then the corresponding closed-form solution can be written as:*

$$\mathbf{Z} = \left((1+\eta)\mathbf{I} - \eta\tilde{\mathbf{A}}\right)^{-1} \tilde{\mathbf{A}}\mathbf{E} \tag{16}$$

*Since $\frac{\eta}{1+\eta} < 1$ for $\forall \eta > 0$, and matrix $\tilde{A}$ has absolute eigenvalues bounded by 1, thus, all its positive powers have bounded operator norm, then the inverse matrix can be decomposed as follows with $k \to \infty$ :*

$$\begin{aligned}
\mathbf{Z} &= \frac{1}{1+\eta} \left( \mathbf{I} - \frac{\eta}{1+\eta}\tilde{\mathbf{A}} \right)^{-1} \tilde{\mathbf{A}}\mathbf{E} \\
&= \frac{1}{1+\eta} \left( \mathbf{I} + \frac{\eta}{1+\eta}\tilde{\mathbf{A}}^1 + \frac{\eta^2}{(1+\eta)^2}\tilde{\mathbf{A}}^2 + \cdots + \frac{\eta}{(1+\eta)^K}\tilde{\mathbf{A}}^K + \cdots \right) \tilde{\mathbf{A}}\mathbf{E} \\
\mathbf{Z} &= \frac{1}{1+\eta}\tilde{\mathbf{A}}\mathbf{E} + \frac{\eta}{(1+\eta)^2}\tilde{\mathbf{A}}^2\mathbf{E} + \cdots + \frac{\eta^{K-1}}{(1+\eta)^K}\tilde{\mathbf{A}}^K\mathbf{E} + \cdots
\end{aligned} \tag{17}$$

*Note that $\frac{1}{1+\eta} + \frac{\eta}{(1+\eta)^2} + \cdots + \frac{\eta^{K-1}}{(1+\eta)^K} + \cdots = 1$ and we can change the coefficient $\eta \in (0, \infty)$ to fit fusion weights $\alpha_1, \alpha_2, \cdots, \alpha_K$. When the layer $K$ is large enough, the propagation mechanism of LightGCN in Equation (3) approximately corresponds to the objective Equation (4).*

---

[*]The corresponding author.  Code: https://github.com/yifeiacc/LogDet4Rec/

## A.2 Proof of Lemma 2

**Proof 2** *First, let us take the gradient of* $\mathrm{tr}\left(\mathbf{Z}^\top \mathbf{L} \mathbf{Z}\right)$ *with respect to the input matrix* $\mathbf{E}$ *and denote* $\mathbf{Z} = \hat{\mathbf{A}}\mathbf{E}$ *where* $\hat{\mathbf{A}} = \sum_{k=1}^K \alpha_k \tilde{\mathbf{A}}^k$.

$$
\begin{aligned}
\frac{\partial \mathcal{L}_{smooth}}{\partial \mathbf{E}} &= \frac{\partial \mathrm{tr}\left(\mathbf{Z}^\top \mathbf{L} \mathbf{Z}\right)}{\partial \mathbf{E}} \\
&= \frac{\partial \mathrm{tr}\left((\hat{\mathbf{A}}\mathbf{E})^\top \mathbf{L}(\hat{\mathbf{A}}\mathbf{E})\right)}{\partial \mathbf{E}} \\
&= 2\hat{\mathbf{A}}^\top \mathbf{L} \hat{\mathbf{A}} \mathbf{E} \\
&= 2\mathbf{Q}\mathbf{E}.
\end{aligned}
\tag{18}
$$

*Treat the weight matrix as a function of the training step* $t$, *i.e.*, $\mathbf{E} = \mathbf{E}(t)$, *then we can derive the gradient of* $\mathbf{E}(t)$ *with respect to* $t$ *by* $\frac{\mathrm{d}\mathbf{E}(t)}{\mathrm{d}t} = 2\mathbf{Q}\mathbf{E}$. *As both* $\mathbf{Q}$ *are fixed, we can solve the equation analytically,*

$$
\mathbf{E}(t) = \exp(2\mathbf{Q}t) \cdot \mathbf{E}(0). \tag{19}
$$

*As we have the non-ascending eigenvalues of* $\mathbf{Q}$ *as* $\lambda_1^{\mathbf{Q}} \geq \lambda_2^{\mathbf{Q}} \geq \cdots \geq \lambda_D^{\mathbf{Q}}$, *we can define an auxiliary function* $f\left(t; \lambda_i^{\mathbf{Q}}, \lambda_j^{\mathbf{Q}}\right) = \exp\left(\lambda_i^{\mathbf{Q}} t\right) / \exp\left(\lambda_j^{\mathbf{Q}} t\right) = e^{(\lambda_i^{\mathbf{Q}} - \lambda_j^{\mathbf{Q}})t}$. *It is obvious that* $f\left(t; \lambda_i^{\mathbf{Q}}, \lambda_j^{\mathbf{Q}}\right)$ *is monotonically decreasing for all* $i > j$. *As* $\mathbf{E}(t)$ *is a transformation of its initial state* $\mathbf{E}(0)$ *up to* $\exp(\mathbf{Q}t)$, *we can conclude that:*

$$
\frac{\sigma_i^{\mathbf{E}}(t)}{\sigma_j^{\mathbf{E}}(t)} \leq \frac{\sigma_i^{\mathbf{E}}(t')}{\sigma_j^{\mathbf{E}}(t')}, \quad \forall t < t' \ \text{and} \ i > j.
$$

*Let the spectrum be following the descending order. Then we have* $\lim_{t\to\infty} f\left(t; \lambda_i^{\mathbf{Q}}, \lambda_j^{\mathbf{Q}}\right) = 0, \forall i > j$ *if* $\lambda_i^{\mathbf{Q}} \neq \lambda_j^{\mathbf{Q}}$.

*Notice the above expression analyses the decay of spectrum for matrix* $\exp(2\mathbf{Q}t)$. *Thus, assume* $\mathbf{E}(0)$ *is a full-rank matrix. Then*

$$
\mathrm{rank}(\exp(2\mathbf{Q}t) \cdot \mathbf{E}(0)) \leq \min\left[\mathrm{rank}(\exp(2\mathbf{Q}t), \mathrm{rank}(\mathbf{E}(0))\right]
$$

*due to the well-known inequality stating that* $\mathrm{rank}(\mathbf{X}\mathbf{Y}) \leq \min\left(\mathrm{rank}(\mathbf{X}), \mathrm{rank}(\mathbf{Y})\right)$.

## A.3 Proof of Corollary 2

Imagine that $\boldsymbol{\Sigma}_{\mathcal{U}} = \mathrm{diag}([1, 0.1])$. Let $\Delta\boldsymbol{\Sigma}_{\mathcal{U}} = \mathrm{diag}([0, -0.1])$ then $\log\det(\boldsymbol{\Sigma}_{\mathcal{U}} + \boldsymbol{\Sigma}_{\mathcal{U}}) = \log\lambda_1 + \log\lambda_2 = \log 1 + \log 0 = -\infty$ so $\mathcal{L}_{logdet} = \infty$. In contrast, for $\mathcal{L}_{soft}$ we have $(\lambda_1 - 1)^2 + (\lambda_2 - 1)^2 = 0.81$. If 10 user feature vectors $f(u) = diag([1, 0.31])$ are in relation with 10 item feature vectors $f(i) = \mathrm{diag}([1, 0.0])$, it is easy to see that $10 \cdot 0.31^2 \approx 1$ which means the alignment loss is better off with the dimensional collapse for $\mathcal{L}_{soft}$ as $1 > 0.81$. For the LogDet penalty however we have $1 \ll \infty$.

## A.4 Proof of Lemma 3

We have the following:

**Proof 3**

$$
D_{\phi_{ld}}(\mathbf{X}, \mathbf{I}) = \mathrm{tr}(\mathbf{X}) - \log\det(\mathbf{X}) - d = \sum_{i=1}^d (\lambda_i - \log\lambda_i - 1). \tag{20}
$$

*Now,* $x - \log x \geq 1$ *with equality at* $x = 1$. *Also,* $x - \log x \geq \log x + 1 - \log 4$ *with equality at* $x = 2$. *Letting* $\lambda_1 \geq \lambda_2 \cdots \geq \lambda_d > 0$, *we have:*

$$
\begin{aligned}
D_{\phi_{ld}}(\mathbf{X}, \mathbf{I}) &\geq (\log\lambda_1 + 1 - \log 4) - (\log\lambda_d + 1) \\
\implies \quad \mathrm{Cond}(\mathbf{X}) &\leq 4\exp D_{\phi_{ld}}(\mathbf{X}, \mathbf{I})
\end{aligned}
\tag{21}
$$

*Thus, LogDet yields an upper bound on the condition number.*

## A.5 Detailed Settings

For the general settings, we create the user and item embeddings with the Xavier initialization of dimension 64; we use Adam to optimize all the models with the learning rate 0.001; the $l_2$ regularization coefficient $10^{-4}$ and the batch size 2048 are used, which are common in many papers [15, 43, 42]. In SimGCL and SGL, we empirically set the temperature $\tau = 0.2$ as this value is often reported the best choice in papers on CL [43, 40]. An exception is that we let $\tau = 0.15$ for XSimGCL on Yelp2018, which brings a slightly better performance. Note that although the paper of SGL [43] uses Yelp2018 and Alibaba-iFashion as well, we cannot reproduce their results on Alibaba-iFashion with their given hyperparameters under the same experimental setting. So we re-search the hyperparameters of SGL and choose to present our results on this dataset in Table 3.

## A.6 Dataset Statistics

Table 6: Dataset statistics.

| Dataset | #User | #Item | #Feedback | Density |
|---|---|---|---|---|
| **Yelp2018** | 31,668 | 38,048 | 1,561,406 | 0.13% |
| **Amazon-Kindle** | 138,333 | 98,572 | 1,909,965 | 0.014% |
| **Alibaba-iFashion** | 300,000 | 81,614 | 1,607,813 | 0.007% |

## A.7 Effective Rank

**Definition 2 (Effective Rank.)** *Consider matrix $\boldsymbol{X} \in \mathbb{R}^{m \times n}$ whose singular value decomposition is given by $\boldsymbol{X} = \boldsymbol{U} \boldsymbol{\Sigma} \boldsymbol{V}^T$, where $\boldsymbol{\Sigma}$ is a diagonal matrix with singular values $\sigma_1 \geq \sigma_2 \geq \cdots \geq \sigma_Q \geq 0$ with $Q = \min\{m, n\}$. The distribution of singular values is defined as the $\ell_1$-norm normalized form $p_i = \sigma_i / \sum_{k=1}^{Q} |\sigma_k|$. The effective rank of the matrix $\boldsymbol{X}$, denoted as $\mathrm{erank}(\boldsymbol{X})$, is defined as* $\mathrm{erank}(\boldsymbol{X}) = \exp\left(H\left(p_1, p_2, \cdots, p_Q\right)\right)$, *where $H\left(p_1, p_2, \cdots, p_Q\right)$ is the Shannon entropy given by* $H\left(p_1, p_2, \cdots, p_Q\right) = -\sum_{k=1}^{Q} p_k \log p_k$.

## A.8 Comparison of Runtimes with the Riemannian Metric

The main advantage of $D_{\phi_{ld}}$ over the Riemannian metric $D_R$ (such as AIRM [10] and LERM [10]) is its computational speed. To compute $D_{\phi_{ld}}$, only determinants need to be computed, which can be efficiently achieved with Cholesky factorization (for $\boldsymbol{\Sigma}_{\mathcal{U}} \boldsymbol{\Sigma}_{\mathcal{I}}$) at a cost of $\frac{1}{3} d^3$ flops [12]. On the other hand, computing the Riemannian metric requires generalized eigenvalues, which takes around $4d^3$ flops for positive definite matrices. Therefore, in general, $D_{\phi_{ld}}$ is much faster to compute (see Table 7b). This speed advantage becomes even more pronounced when computing gradients. Moreover, backpropagation through the matrix determinant is generally stable whereas generalized eigenvalue decomposition suffers undetermined gradients if two eigenvalues are non-simple (equal values). As shown in Table 7a, computing $\partial D_{\phi_{ld}}$ can be over 100 times faster than $\partial D\phi_R$. This difference can be crucial when using gradient-based algorithms, such as neural networks, that rely on the computation of similarity measure gradients.

Table 7: Runtime computed over 1000 trials (millisecs/trial).

(a) Average times to compute gradients.

| $d$ | $\partial_{\mathbf{X}} D_R(\mathbf{X}, \mathbf{I})$ | $\partial_{\mathbf{X}} D_{\phi_{ld}}(\mathbf{X}, \mathbf{I})$ |
|---|---|---|
| 5 | $0.79815_{\pm 0.0934}$ | $0.036_{\pm 0.009}$ |
| 10 | $2.38341_{\pm 0.2094}$ | $0.058_{\pm 0.021}$ |
| 20 | $7.49365_{\pm 0.5954}$ | $0.110_{\pm 0.013}$ |
| 40 | $24.8942_{\pm 1.1264}$ | $0.270_{\pm 0.047}$ |
| 80 | $99.4825_{\pm 5.1813}$ | $0.921_{\pm 0.028}$ |
| 200 | $698.813_{\pm 39.602}$ | $8.767_{\pm 2.137}$ |
| 500 | $6377.22_{\pm 379.11}$ | $94.83_{\pm 1.195}$ |
| 1000 | $40443.0_{\pm 2827.2}$ | $622.2_{\pm 37.70}$ |

(b) Average times to compute function values.

| $d$ | $D_R(\mathbf{X}, \mathbf{I})$ | $D_{\phi_{ld}}(\mathbf{X}, \mathbf{I})$ |
|---|---|---|
| 5 | $0.025_{\pm 0.012}$ | $0.030_{\pm 0.007}$ |
| 10 | $0.036_{\pm 0.005}$ | $0.040_{\pm 0.009}$ |
| 20 | $0.085_{\pm 0.006}$ | $0.061_{\pm 0.009}$ |
| 40 | $0.270_{\pm 0.332}$ | $0.123_{\pm 0.012}$ |
| 80 | $1.234_{\pm 0.055}$ | $0.393_{\pm 0.050}$ |
| 200 | $8.198_{\pm 0.129}$ | $2.223_{\pm 0.169}$ |
| 500 | $77.311_{\pm 0.568}$ | $22.18_{\pm 1.223}$ |
| 1000 | $492.743_{\pm 15.51}$ | $119.7_{\pm 1.416}$ |

## A.9 Performance Comparison w.r.t. to Different Distance Types

Table 8: Main comparison on different distances. ♣ denotes the Matrix Norm and ♠ denotes the Bregman Matrix Divergence. KL Matrix Div. is Bergman Div. associated with $\phi(\mathbf{X}) = \sum_i \lambda_i \log \lambda_i$.

| Geometries | $D(\boldsymbol{\Sigma}_{\mathcal{X}}, \mathbf{I})$ | Yelp2018 | | iFashion | |
|---|---|---|---|---|---|
| | | Recall@20 | NDCG@20 | Recall@20 | NDCG@20 |
| Euclidean Norm ♣ | $\|\boldsymbol{\Sigma}_{\mathcal{X}} - \mathbf{I}\|_2$ | 0.0563 | 0.0459 | 0.0890 | 0.0404 |
| Nuclear Norm ♣ | $\|\boldsymbol{\Sigma}_{\mathcal{X}} - \mathbf{I}\|_*$ | 0.0632 | 0.0516 | 0.0998 | 0.0458 |
| Frobenius Norm ♣♠ | $\|\boldsymbol{\Sigma}_{\mathcal{X}} - \mathbf{I}\|_F$ | 0.0709 | 0.0592 | 0.1112 | 0.0580 |
| KL Matrix Div. ♠ | $\mathrm{tr}\left(\boldsymbol{\Sigma}_{\mathcal{X}} \log \boldsymbol{\Sigma}_{\mathcal{X}}\right)$ | 0.0724 | 0.0602 | 0.1110 | 0.0597 |
| Logdet Div. ♠ | $-\log\det\left(\boldsymbol{\Sigma}_{\mathcal{X}}\right)$ | **0.0732** | **0.0618** | **0.1270** | **0.0617** |

In this section, we adopt other types of distances as in Equation (11) and compare them with proposed method. There are two different categories (1) the matrix norm, *i.e.*, Euclidean Norm, Nuclear Norm, and Forbenius Norm, and (2) the Bregman divergence, *i.e.*, von Neumann divergence (also known as Matrix Kullback–Leibler divergence) and Logdet divergence. The Forbenius norm can belong to both matrix norm and Bregman Matrix divergence. We show their formulas and experimental result in Table 8. We notice that the proposed method (Logdet Div.) achieves the best results.

Table 9: Result for Recall@5, Recall@10, and Recall@20.

| Dataset | Model | Recall@5 | Recall@10 | Recall@20 |
|---|---|---|---|---|
| Yelp2018 | $\text{GCF}_{loget}$ | 0.0275 | 0.0445 | 0.0732 |
| | DirectAU | 0.0255 | 0.0426 | 0.0720 |
| | LightGCN | 0.0211 | 0.0336 | 0.0590 |
| iFashion | $\text{GCF}_{loget}$ | 0.0301 | 0.0483 | 0.0617 |
| | DirectAU | 0.0292 | 0.0493 | 0.0601 |
| | LightGCN | 0.0284 | 0.0391 | 0.0484 |

## A.10 Broader impact and limitations

Our method enjoys impact and limitations similar to those in graph collaborative filtering. Typical GCF models cannot guarantee they can utilize the feature space efficiently. The mode collapse can lead to frequent item bias resulting in a model which is biased in its recommendations. Thus, by improving recommendation of rare items we also offer a generic approach to limiting the recommendation bias which is important in many domains of life. Our method requires no special resources and just a fraction of additional computations beyond what GCF uses. Of course, our model is limited by the GCF model itself and it is applicable only to pipelines that suffer the mode collapse.

