# OpenReview forum: "Mitigating the Popularity Bias of  Graph Collaborative Filtering: A Dimensional Collapse Perspective"
_NeurIPS.cc/2023/Conference — NeurIPS 2023 spotlight_

### Official Review · Reviewer_VmvT · 2023-06-29

**Soundness:** 3 good
**Presentation:** 3 good
**Contribution:** 2 fair
**Rating:** 5
**Confidence:** 3

**Summary:**

The paper discusses the issue of popular bias in Graph Collaborative Filtering (GCF) and demonstrates its resulting Dimensionality Collapse (DC) in the embedding space. The paper points out that LightGCN, widely used in GCF, is prone to DC. The authors introduce a decorrelated penalty based on LogDet divergence to address this problem. Through theoretical analysis and empirical evidence, they demonstrate that GCF LogDet can be extended to full-rank embedding matrices while maintaining a small condition number. On several benchmark datasets, GCF LogDet outperforms other contrastive-based GCF models and improves the recommendation effectiveness for unpopular items.

**Strengths:**

This paper explores the limitations of graph collaborative filtering algorithms in the field of recommendation systems and proposes a novel graph convolutional neural network, GCF-LogDet, which effectively addresses the issues of dimensionality collapse and popularity bias in collaborative filtering. Specifically, the paper demonstrates, from the perspective of the representation space, the problem of dimensionality collapse caused by graph convolution and formulates high-quality self-supervised embeddings in recommendation systems from the perspective of reducing redundancy. Moreover, by using LogDet distance instead of the traditional SoftPlus, the paper successfully avoids the issue of drastic variance changes, providing thorough analysis and discussion, and achieving improved experimental results. Additionally, the paper exhibits high quality and clarity by providing detailed descriptions of the methodology and experimental setup, and it provides a detailed proof of the theorem in the supplementary materials.

**Weaknesses:**

1. In the paper, the title emphasizes alleviating popularity bias, while the analysis and emphasis primarily focus on the issue of dimensionality collapse. However, the direct correlation and equivalence between dimensionality collapse and popularity bias are not explicitly explained or established in the paper. To provide a clearer understanding of the relationship between dimensionality collapse and popularity bias, it would be beneficial for the authors to explicitly discuss and clarify the connection between these two issues in the introduction or related work section. Additionally, the authors should elaborate on how the proposed method specifically addresses the problem of popularity bias, providing a theoretical or empirical explanation of how mitigating dimensionality collapse can effectively alleviate popularity bias in the context of recommendation systems.
2. This paper seems to lack a comprehensive comparison with existing methods that specifically address popularity bias in the context of graph collaborative filtering. As far as I know, [1] method introduces r-regularization based graph aggregation plugin to alleviate the popularity bias during neighbor aggregation of graph convolution. I recommend that the authors revise the experimental section to include broader comparisons, including existing methods that directly address popularity bias (e.g. [1-2]).
[1] Zhao M, Wu L, Liang Y, et al. Investigating Accuracy-Novelty Performance for Graph-based Collaborative Filtering[C]//Proceedings of the 45th International ACM SIGIR Conference on Research and Development in Information Retrieval. 2022: 50-59.
[2] Wei T, Feng F, Chen J, et al. Model-agnostic counterfactual reasoning for eliminating popularity bias in recommender system[C]//Proceedings of the 27th ACM SIGKDD Conference on Knowledge Discovery & Data Mining. 2021: 1791-1800.
3. I suggest that the authors revise the related work and comparative methods sections to include a comprehensive comparison with existing approaches that tackle dimensionality collapse. Specifically, they should discuss and compare methods such as [3-4] that have been proposed to alleviate dimensionality collapse. For example, [3] analyzes the causes of feature collapse from the perspective of graph contrastive learning and proposes a spectral data augmentation method, which is highly relevant to the work presented in this paper.
[3] Zhang Y, Zhu H, Song Z, et al. Spectral Feature Augmentation for Graph Contrastive Learning and Beyond[J]. arXiv e-prints, 2022: arXiv: 2212.01026.(Accpet by AAAI23)
[4] Chen X, He K. Exploring simple siamese representation learning[C]//Proceedings of the IEEE/CVF conference on computer vision and pattern recognition. 2021: 15750-15758.


**Questions:**

1. According to the proof of the paper, there should be a problem of feature collapse in the graph convolutional neural network. Why did the author choose the recommended scene for proof instead of more general graph data?
2. What is the connection between dimensionality collapse and popularity bias, and can they be equated?
3. Why are SimGCL and DirectAU not added to the comparison method in Figure 2 and Figure 4? Obviously, the author's loss function is the closest to DirectAU.


**Limitations:**

1. The authors did not specify the relationship between popularity bias and dimension collapse in detail, which makes readers question the rationality of the author's method.
2. The paper does not compare with the method of alleviating the popularity bias, and it does not reflect the superiority of the method proposed by the author in alleviating the popularity bias problem.

---

> ### Author Rebuttal · Authors · 2023-08-09
>
> # Response to Reviewer 5 (VmvT)
>
> ***We thank the reviewer** for the constructive review and valuable questions that have helped us improve our work.*
>
> ## 1. Title emphasizes alleviating popularity bias but the analysis focus on the dimensionality collapse.
>
> - **Dimensionality collapse (DC) and Popularity bias are two sides of the same coin**: The prevalence of popularity bias in GCF can be attributed to **the significant imbalance within the data distribution across the recommendation bipartite graph**. This distribution often features numerous users engaging with a small subset of items, within a much larger interaction space encompassing millions or billions of items.
> \
> \
> Consequently, during the process of message passing within GCF, a few popular items can disproportionately influence learning, leaving out other items. **This leads to the embedding space where very few user embeddings are around popular item embeddings**.
>
> - **Kindly see Figure 1 (the rebuttal pdf)**. We used Yelp2018 and (top) LightGCN *\vs.* (bottom) our Logdet (also with LightGCN). We applied 2D PCA to embeddings and $\ell_2$ renormalized to the unit sphere.
>  \
>  \
>  The top figure shows that:
>    - the unpopular items clustered in just one cluster
>    - the corresponding user cluster is non-dense so unpopular items do not get recommended often (**popularity bias**)
>    - **the gaps in the unpopular item embedding** cause `effective rank' drop: we refer to it as **the dimensional collapse**
>    - **if only the embedding was spread uniformly on the surface of the hypersphere, it would align with dense user clusters** which would reduce **popularity bias**
>   \
>   \
>   The bottom figure shows that:
>    - **the Logdet managed to spread unpopular user embeddings more uniformly** on the surface of hypershpere - this means `larger effective rank' and thus reduced dimensional collapse**
>   - thanks to more uniformly spread unpopular items, now they formed 3 clusters, and 2 of them align well with dense user clusters, which means **the popularity bias has been reduced**
> \
> \
> **This real-data experiment explains the connection between the popularity bias and the dimensional collapse (understood as reduced `effective rank')**, and in **Fig. 3 (main paper) we showed that Logdet achieves the best effective rank**.
>
> ## 2. Relation/equivalence of DC and popularity bias not explicitly explained/established.
>
> - **the above experiment explains the connection of DC to popularity bias and the effective rank** (very interesting experiment)
> - moreover, consider the alignment with uniformity loss (Eq. 6, main paper). Calculate the gradient the $\nabla_{f_\theta(i)} L$:
> $\nabla_{f_\theta(i)} L= \underbrace{-2\mathbb{E}\_{(u, i)\sim\mathcal{R}} (f\_\theta(u)-f\_\theta(i))}\_{\text{term 1}}+\underbrace{\mathbb{E}\_{i^-\sim\mathcal{I}}\frac{-2f\_\theta(i^-)e^{\left(2-2 f\_\theta(i)^{\top} f\_\theta(i^-)\right)}}{\omega}}\_{\text{term 2}}$ where $\omega$ is a const.
> \
> \
> If $f_\theta(i)$ is popular, **term 1 dominates** the loss and the gradient, clustering users and items together into dense cluster (as in Fig. 1). If $f_\theta(i)$ is unpopular, **term 1** at best aligns few users and unpopular items. **Term 2** may try spread a bit item embedding but it has no dominating effect (to few iterations, finite penalties).
> \
> \
> In contrast, **our logdet loss encourages covariances of users and items to be isotropic** and forces them to be **full rank**. Dense unpopular item cluster is decompacted and spread around the surface of hypersphere. User embedding is also  spread better around the surface of hypersphere. **Effective ranks are improved for user and unpopular items (Fig. 1 in the rebuttal PDF) and so the popularity bias is mitigated.**
>
> ## 3. Compare with existing methods that specifically address popularity bias.
> - Thank you for pointing out following works:
> > [1] *Investigating Accuracy-Novelty Performance for Graph-based Collaborative Filtering*, ACM SIGIR'22.
> \
> > [2] *Model-agnostic counterfactual reasoning for eliminating popularity bias in recommender system*, ACM SIGKDD'21.
>
>   We have now cited them/added experiments. We chose Yelp2018 (common on both papers and us). We selected LightGCN + $r$-adj\_norm and LightGCN+MACR:
>
> |Yelp2018|Recall@20|NDCG@20|
> |-|-|-|
> |LightGCN + Logdet (ours)|**0.0720**|**0.0607**|
> |LightGCN + $r$-adj\_norm|0.0593|0.0485|
> |LightGCN+MACR|0.0631|0.0519|
>
> ## 4. Revise the related work/compare with approaches that tackle DC.
>
> We have now revised/added experiments on:
> - *Spectral Feature Augmentation...* (AAAI'23)
> - *Exploring simple siamese representation learning* (CVPR'21)
>
> We have implemented them in the typical GCF setting (two-layer LightGCN+BPR loss).
>   - For SFA, we added SFA after the LightGCN layer.
>   - For Siamese, we followed SGL but changed the symmetric branch to  unsymmetric and applied the stop gradient:
>
> |Yelp2018|Recall@20|NDCG@20|
> |-|-|-|
> |Logdet|**0.0720**|**0.0607**|
> |Siamese| 0.0602|0.0494|
> |SFA|0.0651|0.0579|
>
> SFA is suboptimal for recommenders as it only partially balances the spectrum (incomplete power iter.) and adds noise (good for node classification but bad for user-item alignment).
>
> ## 5. Why recommender not general graph?
>
> We are interested in problems as in Eq. 13, where user and item embeddings must be aligned:
> * In contrast to general graph data, recommendation datasets have highly skewed data distribution over the bipartite graph (most users interact sporadically with a very small set of items within a vast interaction space of millions/billions of items)
> * Both embeddings of the user and item may collapse. **When users collapse while items are uniformly distributed**, items located in the dense region of the user will be frequently recommended while items in sparse areas will be ignored. **When an item collapses while users are uniformly distributed**, clustered Items are uniformly recommended to the random user.

---

> > ### Comment · Area_Chair_LS5u · 2023-08-13
> >
> > Dear reviewer VmvT, please take your time to carefully review the author's rebuttal and provide your response. Thank you!

---

> > ### Comment · Reviewer_VmvT · 2023-08-18
> >
> > The authors address my main concern, I appreciate the authors' efforts and answers, raising my rating to 5.

---

> > > ### Author Response · Authors · 2023-08-18
> > >
> > > Thank you. We are very happy that we were able to address your main concerns.

---

### Official Review · Reviewer_bxqe · 2023-07-06

**Soundness:** 3 good
**Presentation:** 3 good
**Contribution:** 3 good
**Rating:** 7
**Confidence:** 2

**Summary:**

The paper proposes a novel graph collaborative filtering (CF) approach which tackles the issue of item popularity bias. Starting from the (theoretically-proven) assumption that state-of-the-art graph CF architectures (such as LightGCN) may enhance the so-called dimensional collapse (DC) problem, the authors decide to leverage decorrelation of the learned embeddings through the LogDet divergence. Extensive experiments on three popular recommendation datasets, against multiple state-of-the-art (graph)-based recommendation approaches demonstrate the efficacy of the proposed solution. Additionally, the authors show that the model is able to outperform other graph CF contrastive solutions which optimize uniformity. Finally, a quantitative analysis proves how the approach can tackle the dimensional collapse issue and address the item popularity bias.

**Strengths:**

+ The proposed idea is sound and well-presented.
+ The solution is well-placed in the related literature.
+ The experiments are extensive with multiple analyzed dimensions.
+ Codes and datasets are shared.

**Weaknesses:**

To the best of my knowledge, I cannot see any particular issues.

**Questions:**

To the best of my knowledge, I cannot think about any specific questions.

**Limitations:**

To the best of my knowledge, I cannot see any particular limitations.

---

> ### Author Rebuttal · Authors · 2023-08-09
>
> # Response to Reviewer 4 (bxqe)
>
> ***We thank the reviewer** for the constructive review and valuable questions that have helped us improve our work.*
>
> To further convince reviewer of the value of our work, **we have added Figure 1 in our rebuttal PDF.** The occurrence of dimensional collapse is closely related to the presence of popularity bias, as depicted in **Figure 1 of our rebuttal PDF**. The figure reveals that when the feature vectors of unpopular items are not spread on the surface of $\ell\_2$ ball well, the embedding may cluster in such a way that the corresponding user cluster is non-dense (so this impedes recommendation). The gaps in the embedding of unpopular items represent what we refer to as the dimensional collapse (the `effective rank' of covariance of such embedding is lesser than the well-conditioned case in the bottom Figure 1.) Figure 1 has been prepared with 2D PCA over actual embedding on Yelp2018 for LightGCN and our Logdet (also based on LightGCN).
>
> | Yelp2018 | Amazon-Kindle | iFashion |
> |-|-|-|
> | NCF      | 0.573    | 0.1857  |  0.0831  |
> | LightGCN | 0.0591   | 0.1871  |  0.0845  |
> | LightGCN + $\mathcal{L}\_{soft}$    | 0.0672   | 0.1975  |  0.0972  |
> | LightGCN + $\mathcal{L}\_{logdet}$  | **0.0694**   | **0.2103**   | **0.1110** |

---

> > ### Comment · Reviewer_bxqe · 2023-08-18
> >
> > I carefully read the other reviews and the authors' responses. I would like to thank the other reviewers and the authors for the fruitful discussion so far.
> >
> > I was already quite convinced on the acceptance of the paper, and what I read in the authors' rebuttal further convinced me. I decide to keep my score as it was assigned in the first place.

---

> > > ### Author Response · Authors · 2023-08-18
> > >
> > > Thank you. We appreciate your response, and we are glad that our rebuttal is reassuring.

---

### Official Review · Reviewer_Qs55 · 2023-07-07

**Soundness:** 4 excellent
**Presentation:** 3 good
**Contribution:** 4 excellent
**Rating:** 8
**Confidence:** 3

**Summary:**

In this paper, the authors study the dimension collapse problem in graph-based recommenders. The authors provide both sufficient experimental and theoretical analysis on the occurrence of dimension collapse to popular items and propose a new loss function to encourage feature diversity. Experiments are also sufficient to prove the advantages of the proposed methods. In summary, I think it is a good paper with both sound experiments and theoretical analysis. I suggest a strong acceptance.

**Strengths:**

1. The paper provides both theoretical and experimental analysis of the dimensions collapse problem in graph collaborative filtering

2. The proposed method is simple and effective, boosting the performance of direct representation learning for collaborative filtering learning.

3. Experiments are solid, and also sufficient to validate all the advantages of the proposed method.


**Weaknesses:**

I do not think this paper has obvious weaknesses to cope with


**Questions:**

Do authors think the dimension collapse problem only existed in graph-based collaborative filtering? How about matrix factorization?

**Limitations:**

I do not think this paper has obvious limitations

---

> ### Author Rebuttal · Authors · 2023-08-09
>
> # Response to Reviewer 3 (Qs55)
>
> ***We thank the reviewer** for the constructive review and valuable questions that have helped us improve our work.*
>
> ## 1. Does the dimension collapse exist only in graph-based collaborative filtering? How about matrix factorization?
>
> - The phenomenon of dimensional collapse is not limited to GCF, it also extends to matrix factorization-based approaches.
>
> - The occurrence of dimensional collapse is closely related to the presence of popularity bias, as depicted in **Figure 1 of our rebuttal PDF**. The figure reveals that when the feature vectors of unpopular items are not spread on the surface of $\ell\_2$ ball well, the embedding may cluster in such a way that the corresponding user cluster is non-dense (so this impedes recommendation). The gaps in the embedding of unpopular items represent what we refer to as the dimensional collapse (the `effective rank' of covariance of such embedding is lesser than the well-conditioned case in the bottom Figure 1.) Figure 1 has been prepared with 2D PCA over actual embedding on Yelp2018 for LightGCN and our Logdet (also based on LightGCN).
>
> - **We further demonstrated the occurrence of dimensional collapse in matrix factorization-based techniques**. Specifically, we trained the Neural Collaborative Filtering (NCF) and LightGCN models, monitoring the effective rank of the user embedding matrix as outlined in Section A.7, Definition 2, which can be found in the appendix of the main paper. **The outcomes are depicted in Figure 2(c) of the rebuttal PDF**.
>
> - A lower effective rank signifies a reduced occurrence of dimensional collapse. As we demonstrated in the main paper, LightGCN is particularly susceptible to dimensional collapse, NCF has consistently lower effective rank, revealing that even this matrix factorization-based approaches suffers from dimensional collapse.

---

> > ### Comment · Reviewer_Qs55 · 2023-08-14
> > **Already strong accept, keep score unchanged**
> >
> > Thank you for authors to address my concern. As I already give a strong accept, I will keep my score unchanged

---

> > > ### Author Response · Authors · 2023-08-15
> > >
> > > Thank you for your kind response, comments and support for our work.

---

### Official Review · Reviewer_CD6b · 2023-07-08

**Soundness:** 3 good
**Presentation:** 3 good
**Contribution:** 3 good
**Rating:** 6
**Confidence:** 3

**Summary:**

This paper studies popularity bias in graph collaboration filtering through dimensional collapse analysis of the embeddings. The authors proposed the LogDet divergence based de-correlation penalty for GCF to improve the performance.

**Strengths:**

This work has thorough analysis of dimensional collapse in GCF and its relationship towards propagation and uniformity. De-correlation enhanced objective for mitigate dimensional collapse in GCF. Proposed LogDet divergence to preserve the range4 space of the covariance matrix.

**Weaknesses:**


* While popularity bias adversely impacts the recommendation quality, the quality of the items may also play a role from the end to end recommendation quality. The paper shows the solid improvement on Recall; considering that nDCG@K does not penalize for missing items, it is a bit hard to tell the overall performance.

* The model names seem inconsistent it the experimental study section, such as that in Figure 4 and Tables 2 & 3, which causes some confusion when reading the sections.

* Some details of the hyperparameter setting on the validation dataset should be presented if it is not trivial for the experimental results.

* While it is good to see the proposed method improves the ability to recommend unpolular items, which potentially helps the diversity, the baseline selection is a bit unclear. According to the numbers in Table 3, the baselines in Figure 4 are not the most compelling ones.

**Questions:**

* A minor suggestion on Figure 3, where the last subfig is used for showing the change of condition number over training steps of just one of the three datasets used in the other subfigs. Why not to plot all for completeness, if really the same trends of condition numbers were observed for all?

* How is the impact of K in NDCG@K in the experiment? Is it selective in the experiment, which might imply the missing items in top-K?

---

> ### Author Rebuttal · Authors · 2023-08-09
>
> # Response to Reviewer 2 (CD6b)
>
> ***We thank the reviewer** for the constructive review and valuable questions that have helped us improve our work.*
>
> ## 1. The model names seem inconsistent it the experimental study (Fig. 4 and Tables 2-3).
>
> We apologize. We have now revised our manuscript accordingly, e.g., we call now our method GCF$\_{logdet}$ in Fig. 4 instead of DEC. We have now made it clear that our evaluations are mainly based on LightGCN and the logdet penalty. New figure is updated as Figure 2a in rebuttal pdf.
>
>
> ## 2. In Fig. 3 (last sub-figure) is showing the change of condition number over training steps of just one datasets.
>
> Thank you. Kindly see **Fig. 2b in the rebuttal PDF.** We have now added the figures for all three datasets. In all three cases, the condition number decreases rapidly as the number of training steps increases.
>
>
> ## 3. While popularity bias adversely impacts the recommendation quality, the quality of items may also play a role in the  recommendation quality. The paper shows the solid improvement on Recall.
>
> Since we focus on the ranking ability of our model in the recommendation, Recall and NDCG are the most relevant metrics and they are standard across various works, e.g.:
> > Yu et al. Are graph augmentations necessary? simple graph contrastive learning for recommendation, SIGIR 22.
> > Wu et al.Self-supervised graph learning for recommendation. SIGIR 23
>
> To further address the reviewer's concern about the overall performance, **we provide below experiments using AUC.** We used the Yelp2018 dataset and for the alignment loss we used a simple cross-entropy loss instead of bpr loss, as this is a standard practice for works focusing on AUC metrics:
> > Guo et al. DeepFM: A Factorization-Machine based Neural Network for CTR Prediction
>
>  DirectAU uses the alignment and uniformity loss one the LightGCN backbone.
>  \
>  Our GCF uses the LightGCN (also reported in the table).
>  \
>  The table below shows that GCF$\_{logdet}$ outperforms GCF$\_{soft}$, DirectAU, the LightGCN baseline and other methods:
>
> |  |  |  |  |  |  |  |
> |-|-|-|-|-|-|-|
> |Yelp2018: | GCF$\_{logdet}$ | GCF$\_{soft}$ | SimGCL| DirectAU| SGL| LightGCN |
> |AUC: | **0.853**| 0.843 | 0.850| 0.849| 0.845| 0.841|
>
>
> ## 4. What is the impact of K in NDCG@K in the experiment? Is it selective in the experiment, which might imply the missing items in top-K?
>
> Kindly see the results for our GCF$\_{logdet}$  which outperforms GCF$\_{soft}$, DirectAU (uniformity loss) and LightGCN baselines:
>
> | Yelp | Recall@5 | Recall@10 | Recall@20 |
> | - | --- | --- | --- |
> | GCF$_{loget}$   | **0.0275**   |  **0.0445** | **0.0732** |
> | DirectAU      |   0.0255     |    0.0426   |   0.0720   |
> | LightGCN      |   0.0211     |    0.0336   |   0.0590   |
>
> | Yelp | NDCG@5 | NDCG@10 | NDCG@20 |
> | --- | --- | --- | --- |
> |  GCF$_{loget}$  | **0.301** | **0.0483**|  **0.0618** |
> |  DirectAU     |   0.292   |   0.0493  |    0.0601   |
> |  LightGCN     |   0.284   |   0.0391  |    0.0484   |
>
> ## 5. Hyperparameter details.
>
> We detail the chosen hyper-parameter as follows:
> |Hyperparameter| Value|
> | --- | --- |
> |embedding size  |64    |
> |trainning epoch |50    |
> |batch size      |2048  |
> |learn rate      |0.001 |
> |$\lambda$          |0.5   |
> |LightGCN Layers |2    |
> |L2 reg. weight  |0.0001|
>
> **We will also include all hyper-parameter details in the paper and in the code.**
>
> ## 6. According to the numbers in Table 3, the baselines in Fig. 4 are not the most compelling ones.
>
> Thank you. Kindly see **Fig. 2a in the rebuttal PDF**. We have now **added  SimGCL and DirectAU** accordingly. Our Logdet still performs well and our conclusions hold.

---

> > ### Comment · Reviewer_CD6b · 2023-08-15
> >
> > I would like to thank the authors for the detailed response. It helps clarify the work. Regrading the scores, it remains valid to reflect the quality of the work.

---

> > > ### Author Response · Authors · 2023-08-15
> > >
> > > Thank you for your kind response, comments and support for our work.

---

### Official Review · Reviewer_pKCn · 2023-07-28

**Soundness:** 3 good
**Presentation:** 3 good
**Contribution:** 3 good
**Rating:** 5
**Confidence:** 4

**Summary:**

This paper aims to study the problem that features tend to occupy the embedding space inefficiently. The authors proposes a decorrelation-enhanced GCF objective that promotes feature diversity. LogDet divergence is leveraged to form a new constraint loss.

**Strengths:**

1.  The studied problem of embedding space degradation in graph-based collaborative filtering is interesting and important.
2.  LogDet based decorrelation penalty is developed, and its ability of maintaining the full-rank constraints is theoretically demonstrated.
3.  Experiments show improvement over some baselines.

**Weaknesses:**

1.  The dimensional collapse issue (also known as overcorrelation) has been studied in [1] but not cited.
> Jin W., Liu X., Ma Y., Aggarwal C. and Tang J. Feature over correlation in deep graph neural networks: a new perspective. KDD, 2022.
2. This issue occurs when stacking too many layers whether or not there is a popularity bias.
3. I think 'logdet' regularisation works so well because it encourages embeddings to become rich in different dimensions, rather than mitigating popularity bias.
Note that the relative proportions in the three scenarios in Figure 4 are very close.

**Questions:**

1. The title mentioned "Mitigating the Popularity Bias". However, it seems not to be the main aim of this paper.
2. Some details are missing. How to implement f_{\theta}.
3. Typos. E.g., migrate -> mitigate

---

> ### Author Rebuttal · Authors · 2023-08-09
>
> # Response to Reviewer 1 (pKCn)
>
> ***We thank the reviewer** for the constructive review and valuable questions that have helped us improve our work.*
>
> ## 1. The dimensional collapse issue (also known as overcorrelation) has been studied in [1] but not cited.
> - Thank you for pointing out this related work. We have now included it in the related works. **The *'Explicit Feature Dimension Decorrelation'* introduced in [1] matches closely with what we refer to as $\mathcal{L}_{soft}$ in our paper (see Eq. 9)**. This technique involves the decorrelation of the feature matrix by encouraging a covariance matrix to approach $\mathbf{I}$ through the utilization of the Frobenius Norm. Within our framework, this is just one solution to the case when the function $\phi(\cdot)$ of the Bregman Divergence is defined as $\phi\_F(\mathbf{X}) = ||{\bf X}||_F^2$ (line 187).
> - As highlighted in our discussion within Corollary 2, **it is important to note that $\mathcal{L}_{soft}$ does not inherently ensure the preservation of the range space of the covariance matrix**. Consequently, it does not actively enforce the covariance matrix to attain full rank. Instead, this `soft' approach applies soft constraints that encourage isotropic and full-rank aspects (but cannot strictly enforce the full rank).
> - In Table 1 (main submission), we summarize the characteristics of various regularization techniques and draw comparisons between them. Our approach, denoted as $\mathcal{L}\_{logdet}$, encourages isotropy while simultaneously employing stringent constraint to impose full rank (see Corollary 1) unlike $\mathcal{L}\_{soft}$.  Importantly, **Table 4 (main submission) shows that our $\mathcal{L}\_{logdet}$ outperforms $\mathcal{L}\_{soft}$ in terms of recall@20** on Yelp2018, Amazon-Kindle and iFashion (**7.32%** *vs.* 7.09%, **22.5\%** *vs.* 19.8% and **12.7\%** *vs.* 11.1%).
>
> ## 2. This issue occurs when stacking too many layers whether or not there is a popularity bias.
> - Stacking many layers in graph neural networks can exacerbate popularity bias, leading to an overcorrelation of features. Popularity bias persists even when employing a minimal number of graph neural network layers in GCF, e.g., the commonly used GCF backbone, LightGCN, typically incorporates no more than 3 layers.
>
> **Even with one layer (no over-smoothing)**, we get improved recall@20 with our logdet model compared to soft uniformity penalties, LightGCN, and NCF (MF-based model):
> | Yelp2018 | Amazon-Kindle | iFashion |
> |-|-|-|
> | NCF      | 0.573    | 0.1857  |  0.0831  |
> | LightGCN | 0.0591   | 0.1871  |  0.0845  |
> | GCF$_{soft}$    | 0.0672   | 0.1975  |  0.0972  |
> | GCF$_{logdet}$  | **0.0694**   | **0.2103**   | **0.1110** |
>
> - The prevalence of popularity bias in GCF can be attributed to **the significant imbalance within the data distribution across the recommendation bipartite graph**. This distribution often features numerous users engaging with a small subset of items, within a much larger interaction space encompassing millions, or even billions of items.
>
> Consequently, during the process of message passing within GCF, a few popular items can disproportionately influence the entire learning procedure, leaving out other items with infrequent recommendations. **This dynamics leads to the embedding space wherein the user embeddings are around a select few popular items, resulting in dimensional collapse**. A visual representation of this phenomenon is depicted in **Fig. 1 (the rebuttal pdf)**. Notably, the user and item embeddings are generated using a two-layer LightGCN configuration.
>
> ## 3. Logdet regularisation works so well as it encourages embeddings to be rich in different dimensions, rather than mitigating popularity bias (relative proportions in Fig. 4 are very close).
>
> - The popularity bias is mainly caused by the skew distribution of users which is shown as the dimensional collapse in the latent embedding space.
> - Our method can effectively improve **the performance of recommending the unpopular item (alleviating the popularity basis)**, we add more baselines and show the relative improvement numerically in the table below:
>
> | Yelp2018 | Unpopular | Normal | Popular |
> |-|-|-|-|
> | SGL | 0.630| 0.83| -0.112|
> | SimGCL | 0.521| 1.21|  0.160|
> | DirectAU| 0.331| 0.91|  0.110|
> | GCF$_{logdet}$ | **1.51** | **1.48** | **0.252** |
>
> | Alibaba-iFasion | Unpopular | Normal | Popular |
> |-|-|-|-|
> | SGL | 0.331   | 0.055     | -0.087    |
> | SimGCL | 0.442   | 0.148     | -0.035    |
> | DirectAU| 0.272   | 0.074     | -0.070    |
> | GCF$_{logdet}$   |**0.941**| **0.185** | **0.052** |
>
>
> Kindly notice, our $\mathcal{L}\_{logdet}$ achieves best relative improvements on  Unpopular, Normal and Popular items compared to other methods. However, considering our method alone, we get the biggest relative gain on **Unpopular** items (**0.941**), followed by **Normal** items (**0.185**) and then **Popular** items (**0.052**). Such a trend supports our observation that $\mathcal{L}\_{logdet}$ tackles the popularity bias.
>
> Nonetheless, **we agree that having more isotropic covariance by preventing i.e. rank deficient covariance matrix** has positive impact also on normal and popular items. This is the natural consequence of such a design.
>
> **By dimensional collapse in this paper, we mean lower `effective rank' than possible** which makes embedding space less efficient, yes.
>
> Kindly also **see Fig. 1 in the rebuttal PDF**. We illustrate there that indeed **unpopular items are not distributed uniformly on the surface of hypersphere (which lowers the efficient rank)** for ordinary LightGCN. **After applying the logdet penalty, the spread across the $\ell_2$ ball surface improves**, and infrequent items are more often recommended as a result.
>
> Kindly also refer to a **detailed description of Fig. 1 in Resp. 1 to Reviewer 5 (VmvT).**

---

### Author Rebuttal · Authors · 2023-08-09

**Kindly check the rebuttal PDF for Figures (bottom of this panel)** requested by reviewers. Individual responses across rebuttal refer to them.

---

### Decision · Program_Chairs · 2023-09-21

**Decision:**

Accept (spotlight)

**Comment:**

Overall, all reviewers agreed that the research problem addressed in the submission is of great importance and the experiments conducted are very reliable. Additionally, the authors effectively addressed most of the reviewers' concerns through their rebuttal.